# *WT1* Gene Mutations, rs16754 Variant, and *WT1* Overexpression as Prognostic Factors in Acute Myeloid Leukemia Patients

**DOI:** 10.3390/jcm11071873

**Published:** 2022-03-28

**Authors:** Dorota Koczkodaj, Szymon Zmorzyński, Beata Grygalewicz, Barbara Pieńkowska-Grela, Wojciech Styk, Sylwia Popek-Marciniec, Agata Anna Filip

**Affiliations:** 1Department of Cancer Genetics with Cytogenetic Laboratory, Medical University of Lublin, 20-059 Lublin, Poland; dorotakoczkodaj@umlub.pl (D.K.); sylwia.popek@gmail.com (S.P.-M.); aafilip@hotmail.com (A.A.F.); 2The Maria Sklodowska-Curie National Research Institute of Oncology, 02-781 Warszawa, Poland; grygalewicz@yahoo.com (B.G.); barbara.pienkowska-grela@coi.pl (B.P.-G.); 3Department of Psychology, Institute of Pedagogy and Psychology, Warsaw Management University, 03-772 Warszawa, Poland; wojciech.styk@gmail.com

**Keywords:** acute myeloid leukemia, *WT1* gene expression, *WT1* gene mutation, rs16754 variant, chromosomal aberrations, *FLT3* mutation, *NPM1* mutation, *CEBPA* mutation

## Abstract

(1) Background: The aim of our study was the complex assessment of *WT1* variants and their expression in relation to chromosomal changes and molecular prognostic markers in acute myeloid leukemia (AML). It is the first multidimensional study in Polish AML patients; (2) Methods: Bone marrow aspirates of 90 AML patients were used for cell cultures (banding techniques and fluorescence in situ hybridization), and to isolate DNA (*WT1* genotyping, array comparative genomic hybridization), and RNA (*WT1* expression). Peripheral blood samples from 100 healthy blood donors were used to analyze *WT1* rs16754; (3) Results: Allele frequency and distribution of *WT1* variant rs16754 (A;G) did not differ significantly among AML patients and controls. Higher expression of *WT1* gene was observed in AA genotype (of rs16754) in comparison with GA or GG genotypes—10,556.7 vs. 25,836.5 copies (*p* = 0.01), respectively. *WT1* mutations were more frequent in AML patients under 65 years of age (*p* < 0.0001) and affected relapse-free survival (RFS). The presence of *NPM1* or *CEBPA* mutations decreased the risk of *WT1* mutation presence, odds ratio (OR) = 0.11, 95% CI 0.02–0.46, *p* = 0.002 or OR = 0.05, 95% CI 0.006–0.46, *p* = 0.002, respectively. We observed significantly higher *WT1* expression in AML CD34+ vs. CD34−, −20,985 vs. 8304 (*p* = 0.039), respectively. The difference in *WT1* expression between patients with normal and abnormal karyotype was statistically insignificant; (4) Conclusions: *WT1* gene expression and its rs16754 variant at diagnosis did not affect AML outcome. *WT1* mutation may affect RFS in AML.

## 1. Introduction

Acute myeloid leukemia (AML) affects the function of the hematopoietic system and is a genetically heterogeneous disease at cytogenetic and molecular levels [1]. AML development includes multistep events associated with point mutations, as well as chromosomal aberrations [2]. Mutations in genes encoding Wilms tumor 1 (*WT1*), nucleophosmin (*NPM1*), FMS-like tyrosine kinase-3 (*FLT3*), and CCAAT/enhancer-binding protein alpha (*CEBPA*) affect the pathogenesis of AML [3]. 

The *WT1* gene (*locus* 11p13) encodes a transcription factor that regulates cell growth and differentiation [4]. It contains 10 exons, of which 4 (7–10) encode a DNA-binding domain [5]. The *WT1* gene may possess both tumor suppressor and oncogenic functions in childhood Wilms tumors and leukemias, respectively [6]. *WT1* gene mutations are found in 6–15% of newly diagnosed AML patients [3]. Hot spots were described in two exons (7 and 9), in which loss of function mutations can occur [7]. The alterations of the *WT1* gene are associated with younger age, as well as the coexistence of *FLT3* and *CEBPA* mutations [7]. The changes in the *WT1* gene sequence may affect its expression [3]. The role of *WT1* mutations in the pathogenesis of AML remains controversial [3]. 

Among many variants present in the *WT1* gene, special attention has been paid to the single nucleotide polymorphism rs16754 located in exon 7 (at nucleotide 1297A>G) [8]. Some studies have suggested that rs16754 can be a negative prognostic factor in AML [3,9]. The prognosis in AML is affected adversely by *WT1* gene overexpression [10,11]. The WT1 protein shows antiapoptotic activity. Moreover, higher *WT1* expression is associated with lower cell differentiation [12]. However, the exact role of the *WT1* gene in AML pathogenesis has not been completely revealed. The highest levels of *WT1* expression were found in the M3 AML subgroup (according to French-American-British, FAB, classification) and were linked to a failure of achieving complete remission (CR), shorter overall survival (OS), and shorter progression-free survival [13,14]. It is possible that some changes in coding or noncoding regions of the *WT1* gene or some chromosomal aberrations, especially in the form of microdeletions or microduplications, may affect *WT1* gene expression. *WT1* gene variants and their expression may affect AML prognosis and outcome [9]. However, the results remain controversial.

Abnormal divisions of leukemic cells are often the consequence of gene mutations whose products affect the signaling pathways associated with cell proliferation. The prognosis for AML patients is affected not only by *WT1* gene mutation or abnormal expression but also by mutations of the *FLT3*, *NPM1*, and *CEBPA* genes which were also evaluated [15,16].

Most chromosomal aberrations implicate the appearance of certain gene rearrangements, which affect not only the pathogenesis of the disease but also the prognosis and response to the treatment. Clonal chromosomal abnormalities are identified in approximately 55% of adult AML patients and are recognized as independent prognostic factors [17]. Approximately 15% of patients with clonal changes have complex karyotype, i.e., three or more chromosomal alterations [18]. Patients with t(8;21), inv(16)/t(16;16), or t(15;17) have a favorable prognosis. The prognosis is poor for AML patients with complex karyotype or with the aberrations including inv(3)/t(3;3), t(6;9), monosomy 5 or del(5q), and monosomy 7 or del(7q) [19]. The cases of cytogenetically normal AML (CN-AML) are quite heterogeneous and, depending on cryptic molecular changes, are currently classified in the group of intermediate prognoses [20].

The aim of our study was the complex assessment of the *WT1* rs16754 variant and *WT1* mutations and expression in relation to chromosomal changes and molecular markers such as mutations in *FLT3*, *NPM1*, and *CEBPA* genes. Taking into account some of the controversies among current reports and conflicting results, we decided to analyze *WT1* variants in the form of mutations and single nucleotide polymorphism, as well as *WT1* gene expression in AML. Such multidimensional analysis has not yet been carried out among AML patients in the Polish population. 

## 2. Materials and Methods

The study group consisted of 90 newly-diagnosed AML patients (42 women and 48 men) aged between 18 and 85 years, hospitalized at the Department of Hematooncology and Bone Marrow Transplantation, at the Medical University of Lublin between the years of 2008–2020 (Table 1). Control samples were of peripheral blood taken from 100 healthy blood donors at the Regional Blood Center of Kielce, Poland.

The preliminary diagnosis of AML was based on standard FAB (French-American-British) criteria [11]. On the basis of genetic changes detected at diagnosis, patients were stratified into one of four risk categories: favorable, intermediate-I, intermediate-II, and adverse, according to European Leukemia Net criteria [20]. The mean time of follow-up was 27 months.

With informed consent in accordance with the Declaration of Helsinki of 1975, revised in 2008, and after approval from the Medical University Ethics Committee (app. no.: KE-254/24/2011 and KE-0254/229/2013), the bone marrow aspirates were collected to heparin and EDTA collection tubes (Sarstedt, Nümbrecht, Germany), to be used for cell culture and for DNA, RNA isolation, respectively. 

The treatment of patients generally depended on their age, performance status, and medical history. Patients aged 60 years or lower received induction regimen DAC (daunorubicin 60 mg/m^2^ on days 1 through 3, continuous infusion of cytarabine 200 mg/m^2^ on days 1 through 7, and cladribine 5 mg/m^2^ on days 1 through 5). Consolidation treatment consisted of one course of cytarabine 1.5 g/m^2^ on days 1 through 3 with mitoxantrone 10 mg/m^2^ on days 3 through 5 and one course of high-dose cytarabine 2 g/m^2^ twice daily on days 1, 3, and 5. Then those with favorable karyotype or not eligible for hematopoietic stem cell transplantation (HSCT) were treated for about two years with maintenance therapy based on cytarabine combined with daunorubicin or thioguanine; patients with intermediate or adverse karyotype were referred for allogeneic HSCT (allo-HSCT) or auto-HSCT in cases not eligible for allo-HSCT or lacking suitable donor. Patients aged over 60 years of age in good general condition and without significant comorbidities were treated with reduced induction regimen DA (daunorubicin 45 mg/m^2^ days 1 through 2 or 3 and continuous infusion of cytarabine 100 mg/m^2^ on days 1 through 5 or 7); consolidation and maintenance chemotherapy in this group was reduced as compared with younger patients. Finally, patients over 75 years or younger with significant comorbidities or in poor performance status received low-dose cytarabine (20 mg twice daily for 10 days) repeated every 4 weeks until disease progression. Another treatment option in this group of patients is an azacytidine-a hypomethylating agent used at a dose of 75 mg/m^2^ once a day for 7 days during a 28-day cycle. This treatment is continued until unacceptable toxicity or disease progression. Refractory and relapsed patients were treated with different regimens such as ICE (idarubicin, cytarabine, etoposide), CLAG (cladribine, cytarabine, G-CSF), or MEC (mitoxantrone, etoposide, cytarabine). Patients with promyelocytic leukemia received therapy based on idarubicin and ATRA (all-trans-retinoic acid).

### 2.1. Classical Cytogenetics and Fluorescence In Situ Hybridization (FISH)

Unstimulated cultures of both leukemic (from the first portion of the bone marrow aspirates) and normal (peripheral blood of healthy donors) cells were performed in 15 mL of growth medium (RPMI 1640 medium supplemented with 15% FCS, 100 U/mL penicillin, and 50 μg/mL streptomycin 50 μg/mL, Biomed). The cells were cultured for 24–72 h, and then cultures were terminated conventionally. Metaphase spreads were stained using banding techniques (GTG and RHG bands). Karyotypes were described according to the International System for Human Cytogenetic Nomenclature (ISCN) 2020 [4].

FISH was initially performed on uncultured cells from the diagnostic bone marrow specimens using following probes: 13q14.3, D13S319 and D13S25, 5q (5p15.31 and *EGR1*), 7q (7q22.1 and 7q31), 20q (20q12 and 20q13.2), *EVI1* (3q26) breakapart, *MLL* (11q23), *IGH/CCND1* (14q32.3/11q13), *EGFR* (7p11.2), *EWSR1/ERG* (21q22; 22q12)*, MDM2* (12q14.3-q15), *IGH/BCL2* (14q32.3; 18q21) (all from Cytocell), dual-fusion probe the *PML* (15q22) and *RARA* (17q21.1) *loci*, 21q (D21S259, D21S341, D21S342, 21q22.13–q22.2), 17q23/17p12, TP53/CEP17, D5S23/D5S721 (5p15.2), EGR1 (5q31), CEP7/D7S486 (7q31), D8Z2 (CEP8), *MYC* (8q24.12–24.13), *CDKN2A* (9p21), *MYB* (6q23), CEP9 (9p11–q11), *ETV6* (12p13), CEP15 (D15Z4), CEP17 (D17Z1), D20S108 (20q12), D13S319 (13q14.3)/13q34, *RUNX1* (21q22), dual-fusion probe the *BCR* (22q11.2) and *ABL* (9q34) *loci*, *AML1/ETO* (21q22/8q22) (all from VYSIS, Abbott Laboratories), 1q21/SRD (1p36), hTERT 5p15/5q31, *DEK/NUP214* (6p22; 9q34), *FGFR1*(8p12), *MAF/IGH* (16q23; 14q32), *BCL1/IGH* (11q13; 14q32), 22q11/22q13 (all from KREATECH Diagnostics). FISH was performed according to the manufacturer’s instructions. A total of 200 interphase cells per each sample were examined. The cutoff level for each individual probe was determined based on negative sample analysis and calculated as the mean ± 3SD.

### 2.2. DNA Isolation

DNA isolation from bone marrow aspirates (AML samples) and peripheral blood (control samples) was performed using a commercial kit (Qiagen, Hilden, Germany), according to the procedure provided by the manufacturer. The concentration and quality of isolated DNA were checked spectrophotometrically with NanoDrop 2000 (Thermo Scientific, Waltham, MA, USA). DNA was stored at −20 °C. DNA from bone marrow aspirates was used to analyze chromosomal aberrations (with array CGH method) and *WT1* genotyping (with DNA sequencing method). DNA isolated from peripheral blood was used to analyze the *WT1* rs16754 variant.

### 2.3. Array Comparative Genomic Hybridization (aCGH)

Each step of the aCGH method was performed according to the manufacturer’s (PerkinElmer, Waltham, MA, USA) instructions. The microarrays (Constitutional Chip 4.0) contained human DNA clones (about 100–300 kb long) derived from artificial bacterial chromosomes (BAC probes), which covered the entire human genome. The average resolution of the probes was less than 650 kb.

The process of array hybridization was carried out in HS 400 Pro hybridization station (Tecan, Mennedorf, Switzerland). Array scanning was performed in a ScanArrayGx scanner (PerkinElmer). The OneClick software (PerkinElmer) was used to analyze the results obtained with the aCGH technique. The results were described according to ISCN 2020 [4]. 

### 2.4. WT1 Genotyping—Analysis of WT1 Mutations and rs16754 Variant

The determination of all possible mutations in exons 7 and 9 and rs16754 variant (in exon 7) of the *WT1* gene was performed using direct sequencing. Primer sequences and the procedure of sequencing were performed according to Summers et al., 2007 [21]. A total of 25 μL reaction mixture consisted of 100 ng genomic DNA, 10 µM of each primer, 0.25 mM dNTPs mixture, and 0.31 U of HD polymerase (Clontech) with 1 × PCR reaction buffer (Clontech). PCR reactions with modified thermal conditions were the same for exon 7 and 9 analysis: initial denaturation (98 °C-3 min), 35 cycles (98 °C-20 s, 60 °C-10 s, 72 °C-15 s), final elongation (72 °C-5 min). The PCR reactions were performed in Applied Biosystems 9700 Thermal Cycler. Sequencing PCR was performed using BigDye Terminator v3.1 Cycle Sequencing Kit (Applied Biosystems, Waltham, MA, USA) in a thermal cycler (as previously). The sequencing PCR product was purified using an exterminator kit (A&A Biotechnology, Gdynia, Poland). The sequencing run module was StdSeq50_POP7 in genetic analyzer 3130 (Applied Biosystems). The results were analyzed by use of Applied Biosystems software.

### 2.5. RNA Isolation and WT1 Expression 

RNA isolation was carried out according to the procedure developed by Chomczynski and Sacchi [22]. The concentration of isolated RNA was checked spectrophotometrically using a NanoDrop device (Thermo Scientific). The quality of this nucleic acid was checked during electrophoresis in 2% agarose gel. RNA was stored at −80 °C. 

After RNA isolation, the reverse transcription reaction was performed (A&A Biotechnology set). The real-time PCR reaction (Applied Biosystems 7500 Fast) was carried out on the cDNA (100 ng) template. The analysis of *WT1* gene expression was performed using a *WT1* ProfileQuant (Ipsogen, Marseille, France) kit. In real-time PCR, we used primers and probes with the sequences described by Cillioni et al. [23]. The *ABL* gene was the reference gene. The sequences of the primers and probes for *ABL* were used according to the protocol described by Beillard et al. [24]. In accordance with the LeukemiaNet guidelines, the detection of more than 250 copies of *WT1*/10,000 copies of the control *ABL* gene was defined as *WT1* overexpression [23,25].

### 2.6. Analysis for Other Gene Mutations

*FLT3* and *NPM1* gene mutations were analyzed as previously described [26]. The detection of *CEBPA* gene mutation was according to Benthaus et al. 2008 [27]. 

### 2.7. Statistical Analysis 

The efficacy of induction therapy was assessed using response criteria proposed by European Leukemia Net [20]. Overall survival (OS) was defined as the time from diagnosis to death from any cause. Relapse-free survival (RFS) was defined as time calculated from the achievement of remission until the date of relapse or death from any cause. The probabilities of OS and RFS rates were estimated using the Kaplan–Meier method. Genetic risk groups were compared with respect to these parameters using the log-rank test. In the case of *WT1* rs16754 variant, deviations in genotype frequencies in controls (healthy blood donors) and cases (AML patients) from Hardy–Weinberg equilibrium (HWE) were assessed by Chi-squared test with Yates’s correction for the groups with less than five patients [28]. For 95% confidence interval (CI), we assumed *p* = 0.05 and χ^2^ = 3.84; therefore, if the χ^2^ ≤ 3.84 and the corresponding *p* ≥ 0.05, then the population is in HWE, as described previously by Zmorzynski et al., 2019 [29]. Differences with a *p* value less than 0.05 were considered statistically significant. Statistica software version 12.0 (Statsoft, Tulsa, AK, USA) was used for statistical analysis.

## 3. Results

The presented study included 90 AML patients, 42 males and 48 females, with a median age of 62.63 years. Cytogenetic and molecular analyzes were successful in all the individuals investigated within the study.

### 3.1. WT1 rs16754 Variant

The HWE test showed that the genotypic frequencies of the *WT1* rs16754 variant were not in HWE for AML patients (Table 2). The differences in the genotypic and allelic frequencies of *WT1* between the study and control groups were statistically insignificant (Table 3).

The studied *WT1* variant did not influence the risk of AML. Genotypes AA predominated in the intermediate risk group (33/74). Genotypes AA and GA predominated in the adverse risk group (28/74 and 12/13, respectively). All cases with poor risks had homozygous GG (3/3). The presence of GA and GG genotypes did not affect the risk of *WT1* mutations occurrence (*p* = 0.68). We observed higher expression of *WT1* gene in patients with AA genotype in comparison with those with GA or GG genotypes—10,556.7 vs. 25,836.5 NCN copies (*p* = 0.01), respectively. The AA genotypes were associated with lower PLT numbers in comparison with GA + GG genotypes—78.37 vs. 137.18 (G/L), *p* = 0.02 (Table 4). Survival curves were defined by the Kaplan–Meier method and compared using the log-rank test. Among surviving patients, there were no statistically significant differences in OS and RFS between patients with AA genotype and those with GA or GG genotypes—*p* = 0.302 and *p* = 0.365, respectively (Figure 1 and Figure 2). 

### 3.2. WT1 Mutations

*WT1* mutations were found in 26/90 patients (28.9%). A total of 13 patients had *WT1* mutations affecting exon 7 (14.4%), whereas 12 patients (13.3%) had *WT1* mutations in exon 9. Only one patient had *WT1* mutations in both exon 7 and 9 (1.1%). The total number of *WT1* mutations was 24. We found the following mutations:-In exon 7: c.1375G>A (p.A399V), c.1334C>A (p.R385R), c.1382A>T (p.P401P), c.1389delA (p.N404H); c.1320G>A (p.P381S), c.1314C>T (p.V379I), c.1324indelGTACAAGAG/GTACAAGAGGGTACAAGAG (frameshift variant);-In exon 9: c.1590delC (p.L491X), c.1567G>A (p.R463P), c.1557T>A/C (p.T460S/A).

The most common mutations were substitutions (18/24, 75%). In contrast, deletions were found in 5 cases (5/24, 20.8%). One indel mutation was found (1/24, 4.1%). 

*WT1* mutations were present more frequently in AML patients with age under 65 years (*p* < 0.0001) (Table 4). The presence of *NPM1* or *CEBPA* mutations decreased the risk of mutations occurrence in *WT1* gene-OR = 0.11, 95% CI 0.02–0.46, *p* = 0.002 or OR = 0.05, 95% CI 0.006–0.46, *p* = 0.002, respectively (Table 4). *WT1* mutation, as well as *WT1* rs16754 genotypes, did not affect the OS of AML patients (Figure 1). We observed an association of *WT1* mutations with RFS (Figure 2a). A univariate Cox analysis revealed that patients above 65 years of age had a 2.5-fold risk of death, *p* = <0.01 HR = 2.52 (Table 5).

### 3.3. WT1 Expression

The number of *WT1* gene copies (normalized copy number, NCN) was calculated into 10,000 *ABL* gene copies. A value of >250 *WT1* copies per 10,000 *ABL* copies was seen as a manifestation of overexpression of this *WT1* gene. The *WT1* overexpression was observed in all AML patients at the time of diagnosis. The mean of *WT1* expression was 13,273.12 (range from 1342.9 to 172,093). In patients with normal karyotype and with chromosomal aberrations, we observed mean *WT1* expression 12,747.44 (range 1342.9–172,093) and 13,798.79 (range 1458–105,842.5). The difference in *WT1* expression between patients with normal and abnormal karyotypes was statistically insignificant (Table 4). We observed significantly higher *WT1* expression in AML CD34 positive patients in comparison with AML CD34 negative individuals—20,985 (NCN) vs. 8304 (NCN), *p* = 0.039.

### 3.4. FLT3, NPM1, and CEBPA Mutations

Internal tandem duplication of *FLT3* gene (*FLT3*-ITD) was observed in eight patients (8.8%). In all cases, *FLT3*-ITD change involved only one allele. In six patients with *FLT3*-ITD, complete remission could not be achieved. We did not observe a statistically significant difference between *WT1* gene expression in patients with *FLT3*-ITD and patients with *FLT3* wild type-13,628 vs. 13,238, *p* = 0.98. In patients with *FLT3* mutation, hyperleukocytosis was observed in comparison with those with *FLT3* wild type—49.5 vs. 152 G/L (*p* = 0.002), respectively. In patients who reached CR, a lower WBC number was observed—22.9 vs. 59 (G/L), *p* = 0.012. *FLT3* mutation did not impact CR in AML patients (*p* = 0.81).

*NPM1* mutational analysis revealed a type A mutation in exon 12 (956dupTCTG). This type of mutation was observed in 11 patients. Both *FLT3*-ITD and *NPM1* mutations did not coexist together. The mean number of *WT1* copies in patients with *NPM1* mutation was lower than the median value (7663.26 NCN). However, this difference in mean *WT1* NCN copies between AML patients with or without *NPM1* mutation was statistically insignificant (*p* = 0.71). Similar results were observed in the case of patients with or without *CEBPA* mutation—7356 vs. 13,772 *WT1* NCN (*p* = 0.43), respectively. *CEBPA* mutation coexisted with *NPM1* mutation in three cases. In one AML patient, the presence of both–*CEBPA* mutation and *FLT3*-ITD was observed. A univariate Cox analysis identified *NPM1* mutation impacting RFS of AML patients (*p* = 0.04, HR = 3.46) (Table 5). In multivariate Cox analysis, we did not observe any impact of *WT1* mutation, *WT1* overexpression, or molecular changes in *FLT3*, *NPM1*, and *CEBPA* genes on OS and RFS in AML patients. 

### 3.5. Cytogenetic Aberrations

In our study, by use of conventional cytogenetics and aCGH, genomic imbalances were detected in 45 cases (50%). Owing to the aCGH technique, we detected an abnormal number of chromosomes that included gains and losses, as well as small structural changes in the form of duplications and deletions. 

The aberrations detected with classical cytogenetic techniques included whole chromosomes. These changes were mostly related to the following chromosomes: −17 (11%); −18 (9%); −5 (8%), +8, −15, −16, −21 (7%); −3, +6, −7, −12, −14, −20, −22 (5%). Interstitial and terminal deletions were also detected (del(3q), del(5q), del(7q), del(11q), del(13q), del(17p)), translocations (t(3;15), t(5;7), t(8;16), t(8;21), t(9;22), t(10;12), t(11;11) and t(15;17)), inversions (inv(16)), isochromosomes (i(17q)) and the presence of marker chromosomes.

The chromosomal aberrations revealed by aCGH included small DNA fragments and were predominantly related to the following regions: -losses of (5)(q23q32)—20%;-losses of (7)(p12.3q36.3)—13%;-gains of (8)(q12.1q24.3)—10%;-gains of (11)(q12.2q14.1)—12.2%;-losses of (11)(q22q23.3)—14.4%;-losses of (17)(p13.3p13.1)—16.7%;-losses of (18)(p11.32q23)—11%;-gains of (22)(q12.3q13.2)—5.5%.

By using the aCGH technique, we have confirmed the occurrence of most changes detected by classical cytogenetic methods. In our study, we observed amplification of regions—3q26 (*EVI1* gene *locus*), 8q24 (*C-MYC* gene *locus*), and 11q23 (*MLL* gene *locus*). 

No chromosome X and Y aberrations were found using the microarray CGH method; however, such aberrations were seen in classical cytogenetics tests. In the case of deletions and additions, no statistically significant differences were found between the results obtained by classical cytogenetics (GTG and RHG) and aCGH.

The combination of conventional cytogenetics, FISH, and aCGH enabled us to find favorable karyotype in 13 patients (14.4%), intermediate karyotype in 34 patients (37.7%), and adverse karyotype in 43 patients (47.7%). The median OS was not reached in favorable-risk (26.6 months) and in adverse-risk group (12.8 months). It was reached in intermediate-risk group 39.7 months (*p* = 0.123 favorable vs. intermediate; *p* = 0.001 favorable vs. adverse; *p* = 0.001 intermediate vs. adverse). The median RFS (6.39 months) was not reached in adverse-risk group (1.8 month) in comparison with favorable-risk (10.5 months) and intermediate-risk (8.6 months) groups (*p* = 0.125 favorable vs. intermediate; *p* = 0.002 favorable vs. adverse; *p* = 0.017 intermediate vs. adverse). 

## 4. Discussion

In our study, we analyzed the *WT1* rs16754 variant, *WT1* mutations, and *WT1* gene expression taking into account chromosomal changes and molecular markers such as mutations in *FLT3*, *NPM1*, and *CEBPA* genes. To our knowledge, our investigation is the first multidimensional study including analysis of cryptic chromosomal and molecular changes in Polish AML patients.

### 4.1. WT1 rs16754 Variant

In our study, we found that genotypic frequencies of the *WT1* gene were not in HWE. This suggests an association of *WT1* rs16754 genotypes with the risk of disease development. However, in our study, we did not find the association of analyzed genotypes with AML risk. We cannot exclude, however, that the obtained result was due to sampling error. Moreover, analyzed genotypes did not affect OS or RFS of AML patients. It is consistent with results obtained by Kim et al. in Korean patients with AML [30]. In contrast, Long et al. in their meta-analysis observed that the *WT1* rs16754 variant was associated with better survival of AML patients [9]. They concluded that the *WT1* rs16754 variant was an independent favorable-risk marker [9]. Zhang et al. found that GG genotypes were associated with significantly higher *WT1* expression at mRNA level in comparison with GA and AA genotypes [31]. In our study, we observed similar results. The expression of the *WT1* gene was statistically higher in AML patients with AA genotype. 

### 4.2. WT1 Mutations

We observed a higher frequency of *WT1* mutations in AML patients under 65 years of age. This is in concordance with results obtained by Krauth et al. [32]. In their study of 3157 AML patients, they found that *WT1* mutations were more frequent in younger AML patients and were associated with adverse effects in cytogenetically normal AML patients [32]. In concordance with other studies, we did not find a significant relationship between *WT1* mutations and clinical characteristics, including gender, WBC and PLT count, and Hb concentration [3,33]. We did not find an association of *WT1* mutations with normal karyotype. However, we observed a higher frequency of *WT1* mutations in AML patients with *NPM1* or *CEBPA* genes mutations. More frequent *WT1* mutations were observed by Krauth and coworkers in *CEBPA* mutated AML [32]. In contrast, Hou et al. and Rostami et al. did not observe a relationship between *WT1* mutations and the presence of mutations in genes-*FLT3*, *NPM1*, and *CEBPA* [3,34]. In other research, Toogeh and coworkers showed no correlation of *WT1* mutations with *FLT3*-ITD [35]. In our study, we did not observe an association of *WT1* mutations with *FLT3*-ITD. To assess the prognostic impact of *WT1* mutations, we analyzed them taking into account OS and RFS. We found longer RFS in patients with *WT1* mutation. However, this result is not consistent with previous reports [3,33,36,37]. This may be due to the fact that previous reports were conducted in other populations or as a result of sampling error. 

### 4.3. WT1 Expression

The *WT1* gene shows high expression in AML patients, and its expression at diagnosis may be an adverse predictor of disease outcome [38]. It has been confirmed that increased expression of the *WT1* gene accelerated disease progression [38]. In our study, we observed *WT1* overexpression in 97.7% of newly diagnosed AML patients, which is in agreement with previous studies [3,10,16]. In the presented material, all patients qualified for the study had completed chemotherapy. We observed significantly higher *WT1* expression in AML CD34 positive patients in comparison with AML CD34 negative individuals. Yoon et al. carried out testing of 104 diagnosed AML patients with normal karyotype and showed that increased *WT1* expression in bone marrow was significantly increased in AML patients [39]. We did not find a statistically significant difference in *WT1* expression between patients with normal and abnormal karyotypes. Moreover, we did not observe an association between *WT1* expression and OS or RFS. Results of our research confirm the data obtained by other authors [3,40].

### 4.4. Cytogenetic Analysis

We observed the occurrence of chromosome abnormalities, which were identified by conventional karyotype testing, aCGH, and were confirmed by FISH testing. Without conventional cytogenetics and FISH, we would not be able to detect and resolve balanced aberrations, such as translocations. Similar results were found in numerous studies [19,20]. Our data also support the importance of cytogenetic analysis as a component of the routine diagnostic workup of AML. 

The difference in *WT1* expression between patients with normal and abnormal karyotypes was statistically insignificant. Moreover, we did not observe an association between chromosomal aberrations and the presence of *WT1* mutations.

The median OS was not reached in the favorable-risk group, as well as in the adverse-risk group. In the case of RFS, its median value was not reached in the adverse-risk group only. 

The aberration frequently observed using aCGH was the loss of a long arm fragment of chromosome 5 (including *NPM1* gene *locus*), which is in agreement with other studies [41,42]. Costa et al. and Rucker et al. detected losses of small regions in 5q [43,44]. Mehrotra et al. analyzed 48 AML patients and observed deletions in the 5q33.3 region [45]. Loss of this region was associated with the achievement of CR [45]. Veigaard et al. drew attention to the 5q23.1–q33.3 region, where they found deletions most frequently [46]. In addition, Schoch et al., in testing AML patients with complex karyotype, found deletions of the 5q14–q35 region, as we did in our study [47]. Itzhar et al. specified 5q31.3–q32 regions which were associated with AML development [48]. The deletion of the 5q31 region was associated with the aggressive course of AML, the occurrence of additional chromosomal aberrations, and an unfavorable prognosis [46]. We did not detect microdeletions or microduplications of *loci,* including *WT1*, *FLT3*, and *CEBPA* genes.

### 4.5. Clinical and Molecular Data

*NPM1* mutations in AML patients are associated with the presence of normal karyotype and are rare in patients with genetic aberrations [49]. We did not notice shorter OS or RFS in patients with *NPM1* mutation. The presence of this mutation is favorable and is associated with higher CR rates [4]. In CN-AML patients with *NPM1* mutation, longer RFS is observed in comparison with that observed in patients with wild-type *NPM1* [20]. In our study, we observed *NPM1* gene mutations only in CN-AML. Patients with *NPM1* mutations and lacking *FLT3*-ITD have a better prognosis [50]. The coexistence of *NPM1* and *FLT3*-ITD mutations is a relatively common phenomenon, observed in about 18% of patients with CN-AML [51]. In our material, we did not find the coexistence of these mutations. Many studies have confirmed the correlation between *FLT3*-ITD and an increased number of leukocytes [52]. Our research has also shown that patients with *FLT3*-ITD mutations have increased leukocytosis. The *FLT3*-ITD adversely affects the response to treatment. However, in the presented material, we did not find an association between CR rate and the presence of the *FLT3* mutation.

### 4.6. Limitations

The limitations of our study are associated with the number of AML patients recruited to the study, as well as an automated DNA sequencing method. The sample size was relatively small, in part due to the low incidence of the disease, as well as applied methods such as aCGH. Chromosomal aberrations are factors that need to be taken into account in determining the prognosis of AML patients. Due to their low resolution, the techniques of classical cytogenetics do not detect all alterations; therefore, other methods are used in diagnosing AML, such as PCR and FISH. However, both techniques require prior knowledge of DNA regions where the mutations have occurred. The 90 AML patients sufficed for most analyzes. However, some were not possible as a result of the low frequency of certain molecular changes. In the case of automated sequencing, it would be recommended to use the next-generation sequencing, which has higher resolution and enables analysis of all exons and introns of the *WT1* gene. It is possible that mutations present in other coding regions or in regulatory regions of the *WT1* gene were not detected in our analysis. 

### 4.7. Conclusions

*WT1* gene expression and its rs16754 variant at diagnosis did not affect outcomes in Polish AML patients. *WT1* mutation may affect RFS in AML. In conclusion, we propose a combination of conventional cytogenetics, FISH, and array CGH as necessary tools to unravel the molecular karyotype of AML.

## Figures and Tables

**Figure 1 jcm-11-01873-f001:**
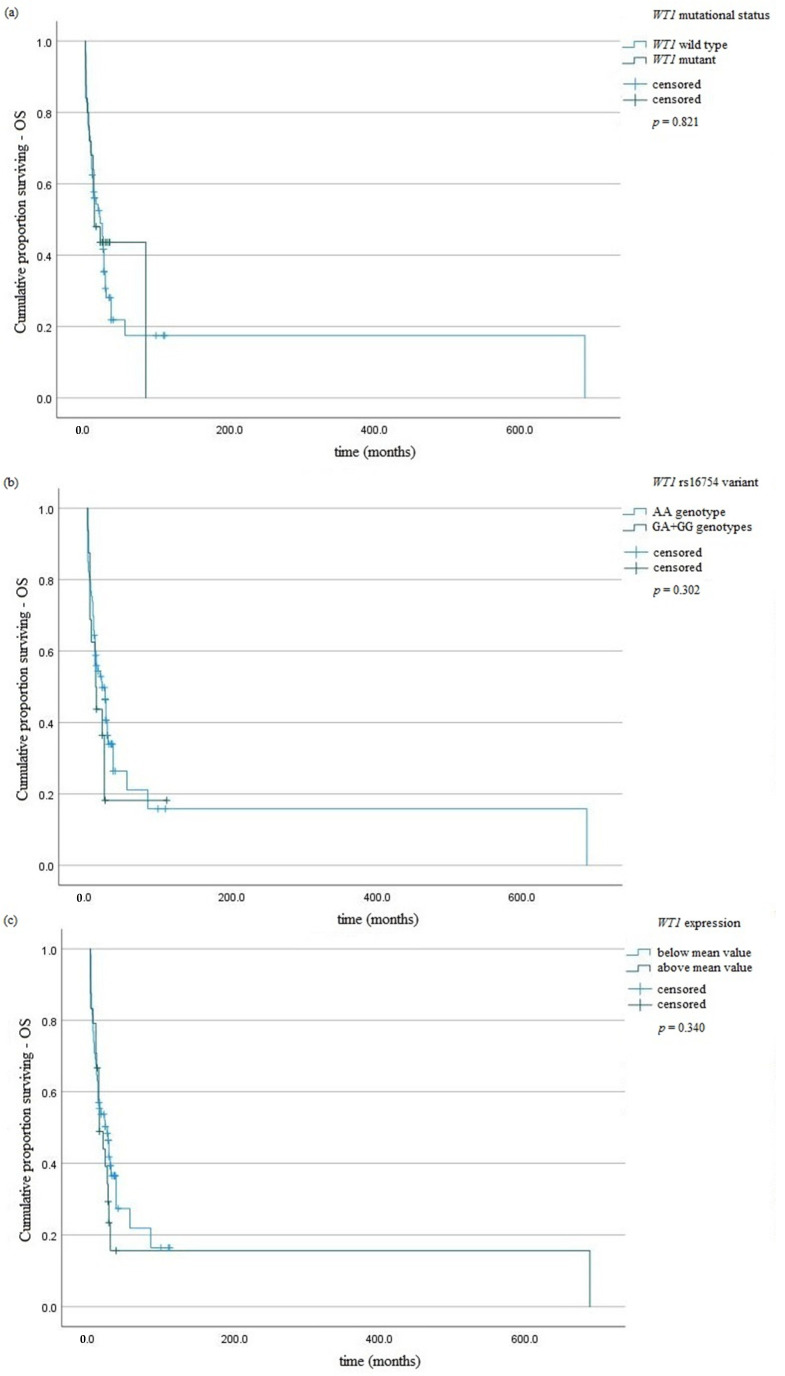
Kaplan–Meier analysis of OS in the group of AML patients taking into account: (**a**) *WT1* mutation; (**b**) *WT1* rs16754 variant; (**c**) *WT1* overexpression.

**Figure 2 jcm-11-01873-f002:**
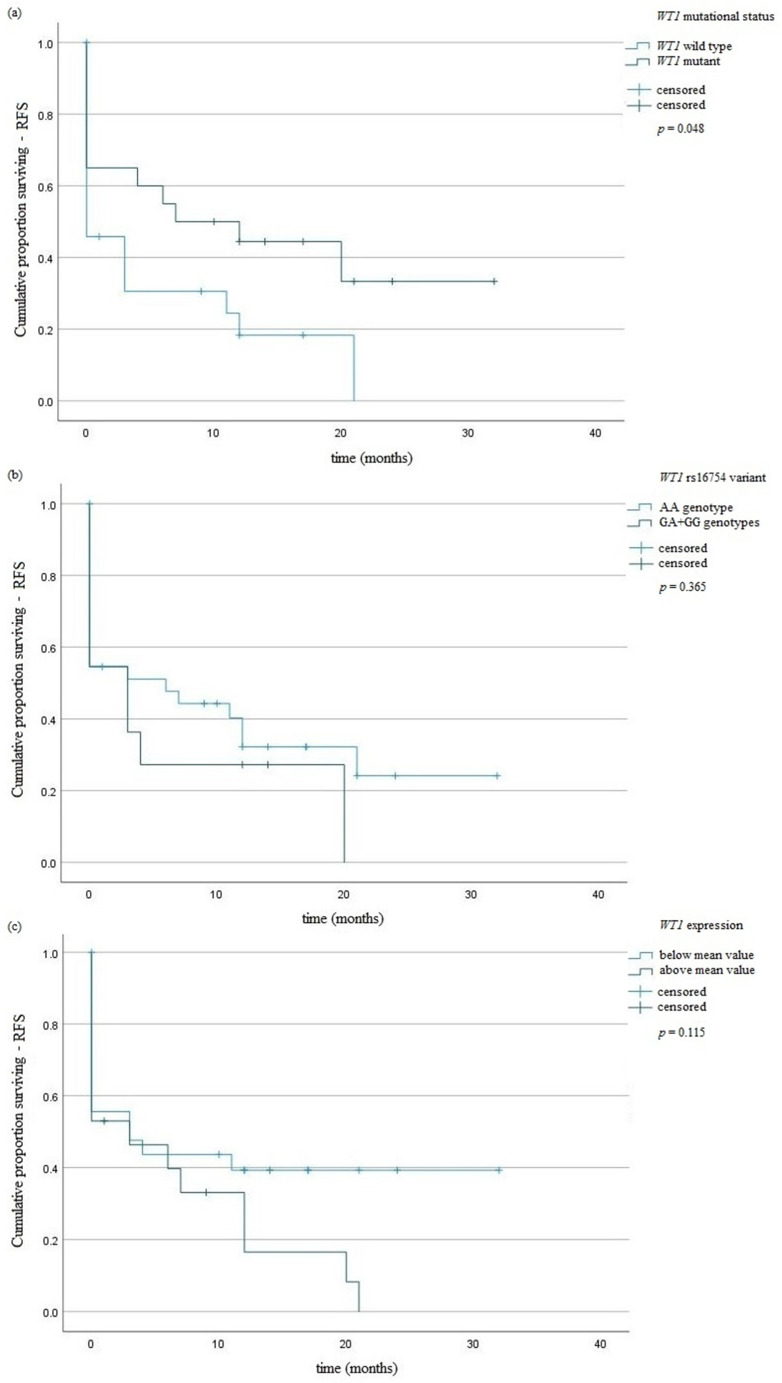
Kaplan–Meier analysis of RFS in the group of AML patients taking into account: (**a**) *WT1* mutation; (**b**) *WT1* rs16754 variant; (**c**) *WT1* overexpression.

**Table 1 jcm-11-01873-t001:** Clinical-laboratory characteristics of patients with AML.

Age median (range)	62.63 (18–85)
Gender (%)	
Male	42 (47)
Female	48 (53)
Laboratory parameters (range)	
HB g/dL	9.26 (4.8–9.7)
WBC G/L	55.15 (0.8–94.84)
PLT G/L	89.06 (6.0–783.0)
FAB subtype:	
M0	6
M1	7
M2	7
M3	12
M4	42
M5	15
M6	1
Risk category:	
favorable	13
intermediate	34
adverse	43
Induction therapy:	
DAC	52
Idarubicin + ATRA	6
AZA	9
reduced DA	15
low-dose cytarabine	8
Stem cell transplantation:	
no	42
allogeneic	37
autologous	11
Cytogenetic:	
Normal karyotype	45
Abnormal karyotype	45 (22 *)
FISH (%):	
del(13)(q14.3)	51 (56.6)
del(17)(p13.1)	15 (16.7)
del(11)(q23)	13 (14.4)
del(6)(q23)	5 (5.5)
t(15;17)(q22;q21.1)	6 (6.6)
*BCR/ABL*	2 (2.2)
*MLL* amplification	1 (1.1)
*MYC* amplification	1 (1.1)
*EVI1* amplification	1 (1.1)
Immunophenotype—negativity/positivity	
CD34	51/39
CD33	19/71
CD14	66/24
Molecular variants—present/absent	
*FLT3*-ITD	8/82
*NPM1* mutation	11/79
*CEBPA* mutation	7/83

* complex karyotype.

**Table 2 jcm-11-01873-t002:** Hardy–Weinberg equilibrium (HWE) for *WT1* rs16754 variant in case and control groups according to expected (E) and observed (O) values.

GROUPS	GENOTYPES			Total	HWE *p* Value and χ^2^ *
*WT1* rs16754 variant					
-	AA	GA	GG	-	-
CONTROL					
E	79.21	19.58	1.21	100	*p* = 0.44, χ^2^ = 0.57
O	80	18	2	100	
CASE					
E	70.2	18.55	1.25	90	*p* = 0.0007, χ^2^ = 11.5
O	74	11	5	90	

* if the χ^2^ ≤ 3.84 and the corresponding *p* ≥ 0.05, then the population is in HWE.

**Table 3 jcm-11-01873-t003:** The comparison of allele frequency and distribution of *WT1* variant among AML patients and controls.

Gene Variants and Alleles	AML n (%)	Controls n (%)	Odds Ratio	95% CI	*p* Values
Codominant model					
AA	74 (82.2%)	80 (80%)	1	-	-
GA	11 (12.2%)	18 (18%)	1.51	0.67–3.41	0.31
GG	5 (5.5%)	2 (2%)	0.37	0.06–1.96	0.41
Dominant model					
AA	74 (82.2%)	80 (80%)	1	-	-
GA + GG	16 (17.7%)	20 (20%)	1.15	0.55–2.40	0.69
Recessive model					
AA + GA	85 (94.4%)	98 (98%)	1	-	-
GG	5 (5.5%)	2 (2%)	0.34	0.06–1.83	0.36
Total:	90 (100%)	100 (100%)			
Alleles					
A	159 (88.3%)	178 (89%)	1	-	-
G	21 (11.7%)	22 (11%)	0.93	0.49–1.76	0.84
Total:	180 (100%)	200 (100%)			

**Table 4 jcm-11-01873-t004:** Correlation of *WT1* variants and overexpression with clinical characteristics in AML patients.

Features	*WT1* AA Genotype	*WT1* GA + GG Genotype	*p* Value	*WT1* Mutated	*WT1* Wild Type	*p* Value	*WT1* Expression *	*WT1* Expression **	*p* Value
Gender									
Male	34	8	0.76	18	38	0.38	37	12	0.61
Female	40	8		8	26		29	12	
Age									
Age < 65 years	62	12	0.63	23	13	<0.0001	55	19	0.64
Age ≥ 65 years	12	4		3	51		11	5	
Cytogenetics									
Normal karyotype	35	10	0.27	16	29	0.16	32	13	0.63
Abnormal karyotype	39	6		10	35		34	11	
Point mutations									
*NPM1* wild type	66	13	0.64	18	61	0.002	58	21	0.75
*NPM1* mutated	8	3		8	3		8	3	
*FLT3* wild type	67	15	0.93	21	61	0.07	62	20	0.25
*FLT3* mutated	7	1		5	3		4	4	
*CEBPA* wild type	69	14	0.80	20	63	0.002	60	23	0.74
*CEBPA* mutated	5	2		6	1		6	1	
Clinical values									
WBC median	42.16	113.59	0.06	13.74	61.19	0.64	50.82	74.63	0.96
Hb median	9.18	9.74	0.33	8.85	9.33	0.27	9.28	9.17	0.17
PLT median	78.37	137.18	0.02	86	90.28	0.19	90.79	84.45	0.45

* *WT1* expression lower than medium value (NCN = 13,273.12). ** *WT1* expression higher than medium value (NCN = 13,273.12).

**Table 5 jcm-11-01873-t005:** Analysis of prognostic impact on OS and RFS in AML patients.

Feature	Univariate Analysis			Multivariate Analysis		
	*p* Value	HR	95% CI	*p* Value	HR	95% CI
OS						
Age	<0.01	2.52	1.33–4,81	0.01	0.41	0.22–0.81
*WT1* mutation	0.82	1.07	0.59–2.33	0.74	0.89	0.48–1.52
*WT1* overexpression	0.35	0.76	0.44–1.34	0.54	0.84	0.45–1.82
*NPM1* mutation	0.13	2.03	0.81–5.10	0.31	1.73	0.61–4.93
*FLT3*-ITD	0.76	0.86	0.35–2.17	0.61	0.78	0.29–1.93
*CEBPA* mutation	0.54	1.44	0.45–4.62	0.96	0.97	0.27–3.27
Abnormal karyotype	0.64	0.89	0.53–1.47	0.77	0.92	0.51–1.56
RFS						
Age	0.07	2.08	0.92–4.65	0.24	0.59	0.27–1.52
*WT1* mutation	0.11	1.84	0.59–3.25	0.76	1.13	0.36–2.18
*WT1* overexpression	0.19	0.63	0.31–1.27	0.59	0.79	0.33–1.83
*NPM1* mutation	0.04	3.46	1.03–6.55	0.17	2.75	0.92–5.45
*FLT3*-ITD	0.58	0.74	0.26–2.13	0.38	0.57	0.15–1.99
*CEBPA* mutation	0.16	2.37	0.70–7.95	0.62	1.42	0.39–5.75
Abnormal karyotype	0.51	0.78	0.37–1.62	0.69	0.84	0.35–1.87

## Data Availability

The clinical data used to support the findings of this study are available from the corresponding author upon request.

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
