# Peer review of "WT1 Gene Mutations, rs16754 Variant, and WT1 Overexpression as Prognostic Factors in Acute Myeloid Leukemia Patients"

_jcm, 2022, doi:10.3390/jcm11071873_

Round 1

Reviewer 1 Report

The manuscript reports a comparison of WT1 expression, variants and mutations in bone marrow samples of 90 naiive AML patients versus peripheral blood of 100 healthy donors.  The study suffers from 3 major limitations due to the samples used, the inclusion of M3 patients, and to the adoption of non-standard therapies (i.e. maintenance chemotherapy for favorable risk patients). Moreover, major determinants of OS such as NPM1 and FLT3 did not recult into a significant HR, which is odd. 

Some syntax errors are widespread in the text.

This topic deserves studies supporting clinical practice recommendations for a proper adoption of WT1 testing. however, the manuscript needs to be improved. 

E.g. The abstract does not clarify all the acronyms used. the abstract does not state how many healthy donors were tested and which samples were used.

Author Response

Dear Reviewer,

thank you for all suggestions and substantive comments to the manuscript. We have taken them into account in the revised version of the manuscript. Below are listed detailed answers to your review report.

1) Limitation associated with samples used.

The study group consisted of bone marrow aspirates collected from 90 consecutive AML patients. Control samples were of peripheral blood taken from 100 healthy blood donors. The sample size was relatively small, in part due to the low incidence of the disease, as well as applied methods – for example array CGH.  The number of 90 AML patients was enough for most analyzes.

2) Inclusion of AML M3 patients and adoption of non standard therapies.

The bone marrow aspirates from AML patients were analyzed by use of array CGH method, which revealed microdeletions and/or microadditions in genetic material. These changes were detected in patients with M3 subtype. Moreover “larger” changes were found in these patients - for example isochromosome of the long arms of chromosome 17 [i(17q)]. The presence of additional changes in karyotypes of M3 patients, changed the prognosis from favorable to intermediate or adverse (in the case of complex karyotypes), hence alternative forms of treatment were used in these patients. It was the reason to include these patients to statistical analysis.

3) Major determinants of OS such as NPM1 and FLT3 did not result into a significant HR, which is odd

No significant HR in univariate and multivariate analyzes might be as a result of low number of AML patients with FLT3 and NPM1 mutations. We found mutations of FLT3 and NPM1 genes in 8 and 11 patients, respectively. These mutations did not coexist together. Moreover, our insignificant results in this field might be as a result of error sampling. In larger AML studies were found associations of NPM1 and/or FLT3 mutations with HR. For example, Hou et al. in a cohort of 470 AML patients found statistically significant lower relative risk in AML patients with NPM1mut/FLT3-ITD- (Hou et al., 2010 - doi: 10.1182/blood-2009-12-259390).

4) Some syntax errors are widespread in the text.

Syntax errors were corrected as suggested by the reviewer.

5) the manuscript needs to be improved.

The manuscript was revised by a native speaker. Moreover, the suggested changes has been made.

6) The abstract does not clarify all the acronyms used. The abstract does not state how many healthy donors were tested and which samples were used.

The abstract was corrected as suggested by the reviewer.

Reviewer 2 Report

Please give a better flow to the manuscript.

Please cite the references appropriately: e.g. line 53 instead of [10],[11] they should be cited as [10, 11].

I do not consider Figure 1, 2, 3 and 4 useful regarding the results of the current manuscript as similar examples are readily available in the literature. Please remove Figure 1, 2, 3 and 4.

Please review Figures 5 and 6 and make them more aesthetically pleasing.

Table 5: Please add the 95% CI in your univariate and multivariable models.

Why did you apply HWE to a genetic variant assessed in tumor cells? It is rather difficult to call a germline variant from a tumor sample.

Overall, the study brings no novelty, it included too few patients and certain aspects of the methodology are questionable as previously mentioned.

Author Response

Dear Reviewer,

thank you for all suggestions and substantive comments to the manuscript. We have taken them into account in the revised version of the manuscript. Below are listed detailed answers to your review report.

1) Please give a better flow to the manuscript.

The manuscript was revised by a native speaker. The suggested changes has been made.

2) Please cite the references appropriately: e.g. line 53 instead of [10],[11] they should be cited as [10, 11].

The literature citations were corrected as suggested by the reviewer.

3) Please remove Figure 1, 2, 3 and 4.

The Figures 1-4 were removed as suggested by the reviewer.

4) Please review Figures 5 and 6 and make them more aesthetically pleasing.

The Figures 5 and 6 were corrected as suggested by the reviewer. Figures 5 and 6 were renumbered and now they have numbers 1 and 2, respectively.

5) Table 5: Please add the 95% CI in your univariate and multivariable models.

The 95% CI  was added in Table 5 as suggested by the reviewer.

6) Why did you apply HWE to a genetic variant assessed in tumor cells?

The Hardy Weinberg equilibrium was assessed to single nulceotide polymorphism rs16754 (g.44143G>A, NG_009272.1 according to NCBI). The rs16754 variant was present in DNA of bone marrow samples from AML patients and peripheral blood of control samples. It suggests that this variant is constitutional. Constitutional single nucleotide polymorphisms are one source of genetic variation and may serve as a means to identify these high-risk individuals (Hahn et al., 2007, doi: 10.2217/14796694.3.6.665). We did not use HWE in the case of WT1 mutations, which were present only in tumor cells.